biophysics/computational physics/statistical physics

microswimmers, control effort, computer simulation, information entropy

**Author for correspondence:**
Julia M. Riede
e-mail: julia.riede@simtech.uni-stuttgart.de

# The control effort to steer self-propelled microswimmers depends on their morphology: comparing symmetric spherical versus asymmetric L-shaped particles

Julia M. Riede[1], Christian Holm[2], Syn Schmitt[1] and Daniel F. B. Haeufle[3]

[1]University of Stuttgart Institute for Modelling and Simulation of Biomechanical Systems, Nobelstraße 15, Stuttgart 70569, Germany
[2]University of Stuttgart Institute for Computational Physics, Stuttgart, Germany
[3]Eberhard Karls Universität Tübingen, Hertie Institute for clinical brain research (HIH) and center for integrative neuroscience (CIN), Tübingen, Germany

JMR, 0000-0002-2453-0618; CH, 0000-0003-2739-310X;
SS, 0000-0002-7768-8961; DFBH, 0000-0002-3480-6892

Active goal-directed motion requires real-time adjustment of control signals depending on the system's status, also known as control. The amount of information that needs to be processed depends on the desired motion and control, and on the system's morphology. The morphology of the system may directly effectuate or support the desired motion. This morphology-based reduction to the neuronal 'control effort' can be quantified by a novel information-entropy-based approach. Here, we apply this novel measure of 'control effort' to active microswimmers of different morphology. Their motion is a combination of directed deterministic and stochastic motion. In spherical microswimmers, the active propulsion leads to linear velocities. Active propulsion of asymmetric L-shaped particles leads to circular or—on tilted substrates—directed motion. Thus, the difference in shape, i.e. the morphology of the particles, directly influence the motion. Here, we quantify how this morphology can be exploited by control schemes for the purpose of steering the particles towards targets. Using computer simulations, we found in both cases a significantly lower control effort for L-shaped particles. However, certain movements can only be achieved by spherical particles. This demonstrates that a suitably designed microswimmer's morphology might be exploited to perform specific tasks.

# 1. Introduction

Active goal-directed motion requires control strategies. Finding a strategy to reach the goal can be essential for survival or reliable cargo delivery [1–4]. Such control strategies need to process information, and, thus, come with a certain *control effort*. The control effort depends not only on the intended motion and the environment [1,5,6] but also on the morphology of the system. In this contribution, we aim to transfer the concept of quantifying control effort from macroscopic to microscopic systems: Our approach to control effort has been developed to quantify muscular contributions in locomotion [7]—we here apply it to microscopic systems, i.e. spherical and *L*-shaped microswimmers. This approach is based on information entropy and explicitly quantifies the information required to achieve a specific movement (control goal) independent of the system's morphology, i.e. the shape of the microswimmers (see §2).

Recently self-propelled spherical microswimmers, e.g. Janus particles, have received a lot of attention. Without active propulsion, these μm-sized spheres show rotational and translational Brownian diffusion in liquids. This random movement is identical in all directions and characterized by the mean square displacement (MSD) law for diffusion described by A. Einstein [8]. Active self-propelled particles overcome natural diffusion by different mechanisms, e.g. diffusiophoresis [9–12]. With constant active propulsion, the short-term translational movement is more deterministic and the MSD characteristics change [9,10,13]. However, the direction is still governed by diffusion and therefore the particle trajectory is still random. To overcome this, external control strategies have been proposed allowing microswimmers to be kept in place or to navigate them to specific targets [5,6,11,12,14–19]. This is achieved by acquiring and processing information on the current state of the particle, e.g. position and orientation, to determine the next suitable action towards the goal. An example: spherical Janus' particles can be navigated towards a target although their propulsion mechanism allows only for unidirectional propulsion. The control strategy is to wait until rotational diffusion randomly orients the particle towards the target. Only then propulsion is activated allowing for a simple navigation of the particle [11,12]. By varying the control strategy, different auxiliary conditions may be considered: minimizing the duration of a movement requires a different strategy than minimizing energy consumption or keeping a particle within a narrow target corridor [15,17].

However, not only does the control strategy influence the behaviour but also the morphology, e.g. the shape. *L*-shaped particles for example show a high degree of asymmetry [20,21], in contrast to spherical Janus particles. With constant active propulsion, the asymmetry of *L*-shaped particles results in a deterministic rotation of the particle in one direction. This rotation is due to the morphology and overlays the random diffusional reorientation. On a horizontal substrate, this results in circular-like trajectories [20]. On a slightly tilted substrate, an even richer set of trajectories emerges in the interaction with gravity (please see figs 1c and 3 in [21]). For low propulsion forces, particles sediment slowly. For medium propulsion, straight trajectories at different angles with respect to gravity emerge. For high forces, loop-like trajectories emerge. Thus, the morphology of the *L*-shaped particles influences the movement. Tuning the propulsion force allows us to tune the trajectory and generate various and to some extent predictable behaviour. Potentially, this could be exploited for specific movement tasks and ultimately reduce their control effort.

So far, it has never been investigated how particles with *L*-shaped morphology could be navigated and for which tasks their particular morphology could be beneficial. From the experimental trajectories reported in [20,21], we expect that (1) the natural rotation of self-propelled *L*-shaped particles could be exploited for circular movements; (2) the rich behaviour in the interaction with gravity in a slightly tilted setup could be exploited for linear movements; (3) that targets exist, which cannot be reached as no control strategy can be found to steer the *L*-shaped particles in their direction.

In this study, we present control strategies for *L*-shaped particles for circular and linear movements. To quantify whether the asymmetric morphology allows us to simplify control for these movement goals, we here apply the information entropy-based measure of control effort [7] to compare *L*-shaped versus spherical (Janus) particles. The relevance of this study is two-fold. On the one hand, we demonstrate the applicability of the measure control effort previously developed for macroscopic deterministic systems [7] to microscopic systems which are governed by stochastic processes. This may be the basis for studying more microscopic systems with interesting (potentially changing) morphology [18,22–24], including biological microorganisms [25–27]. In addition, we show that the interaction between morphology and control strategy can be exploited to simplify control. In this way, measuring control effort may become a relevant benchmark figure of merit in technical applications, e.g. micro assembly or drug delivery, where adequate morphologies may simplify specific tasks.

# 2. Control effort

## 2.1. The concept of control effort

The measure of control effort used in this study to quantify the contribution of morphology to the control has originally been developed for complex macroscopic biological animals. In these tightly integrated systems, it is hard to discriminate the contribution of the neuronal control and the contribution of the morphology, e.g. the visco-elasticity of muscles. However, it had been shown in numerous studies that the morphology may significantly contribute to control (for an overview of this topic, please see [28]). One example is the contribution of the flexibility of bumble-bee wings to the stabilization of the flight [29]. Another example is the nonlinear contraction dynamics of biological muscles, which help to counteract external perturbations during movement without the necessity of neuronal adaptation [30–33].

These studies show that muscles pre-determine the movement to a certain degree—more than e.g. an electric motor would—and therefore the morphology contributes to the control [34]. If this pre-determination is in line with the movement goal, the neuronal effort to control this movement can be reduced [7,35]. More precisely, the amount of information which has to be processed in order to generate the desired movement can be reduced. The minimal information required to perform a certain movement is termed *control effort* and it can be quantified with an information entropy based approach (see §2.2 and [7]).

The key concept to quantify the contribution of morphology to the control is to generate a defined movement with two agents of different morphology. Quantifying the minimally required information, i.e. the control effort, in both agents then shows which of the morphologies is contributing more to the desired movement goal. In the macroscopic biological studies, this was achieved by modelling and computer simulation e.g. of walking. In the model, it is possible to change the morphology by replacing the nonlinear visco-elastic muscles by linear force elements neglecting the muscles' biophysical dynamics [35].

In this study, we transfer this concept to the world of microswimmers. We also simulate defined movements (circular and linear) with two agents of different morphology: *L*-shaped and spherical particles. We implement controllers which steer the particles and generate the defined movements in presence of the stochastic Brownian translation and rotation. Please note that the movements (circular and linear) were explicitly chosen based on previous data which suggested that under specific conditions *L*-shaped particles could or even should perform better. Also, the deterministic contributions in the equations of motion of the *L*-shaped particles shows that these particles should easily generate circular movement and even somewhat linear movements under the very specific conditions of a slightly tilted setup. Hence, the open question remaining was whether the information entropy-based measure of *control effort* would be capable of detecting the expected contribution of the morphology in microswimmers in the presence of a stochastic motion component.

We would also like to emphasize at this point that the more specific morphology comes with a drawback: other goal-directed movements may be harder or even impossible with the *L*-shaped particles. One example is that in a slightly tilted setup, *L*-shaped particles have one dominant direction (in our model towards positive $x$) which makes it impossible to reach targets to the other side (in this case negative $x$ if the particles starts at $x = 0$). This shows that control effort is movement specific and not generally particle specific.

## 2.2. Quantifying control effort

Generating a desired movement, circular or linear, requires navigating the particles. Navigation requires measuring the current position and orientation, processing this information, and adjusting the propulsion accordingly. Simply put: propulsion is only turned on if the particle is oriented towards the movement goal (see §3.3). The amount of information which has to be processed differs between task and particle morphologies. It is mainly determined by the rate at which the data is processed ($1/\Delta t$) and the resolution of the control signals ($\Delta u$). In principle, the coarser the discretization, the less information is processed.

To quantify the information, we resort to Shannon's information entropy [36], which is the basis for our measure of control effort [7]. In a nutshell, we want to quantify the minimum information required to perform a movement. To calculate the information content of the control signal, we need to discretize it. The time is discretized into intervals of $\Delta t$ and the amplitude into intervals of $\Delta u$. The minimum and

maximum signal values $u^{\min}$ and $u^{\max}$ determine the number of possible signal values

$$N = 1 + \frac{u^{\max} - u^{\min}}{\Delta u}.$$

Most of the control strategies switch between zero ($u^{\min} = 0$) and maximum velocity ($u^{\max} = v_0$) depending on the current state of the particle (position and orientation), leading to $N = 2$. However, for the $L$-particles in the linear movement, we allow values in-between, resulting in $N > 2$.

With the assumption of equal distribution of the measurement values [7], the information can be calculated as

$$I = \frac{T_t}{\Delta t} \log_2 \left( 1 + \frac{u^{\max} - u^{\min}}{\Delta u} \right), \tag{2.1}$$

with the time $T_t$ for the particle to reach the target. Here, the information in the signals depends on the time and amplitude resolution. A coarser discretization, meaning larger $\Delta t$ and $\Delta u$, reduces the information content of a signal.

To determine control effort, it is necessary to specify a desired movement and a performance criterion which allows us to quantify the success of the movement. We here investigate two types of movement: circular and linear navigation. We restrict the movement by limiting the allowed region. If the centre of mass leaves the defined region, further referred to as 'target corridor', the attempt is considered a failed navigation. There is no interaction between the particle and a wall, as the corridor is merely a virtual movement constraint. The width of the target corridor is 20 µm in the circular and linear movement. The performance criterion is, thus, the arrival probability ($p_L$, $p_J$, $p_j$ for $L$-particles, large, and small Janus particles, respectively) of particles depending on time and amplitude resolution for a given movement, corridor width, and control strategy. The performance is expected to decrease, i.e. fewer particles arrive, if the time and amplitude intervals ($\Delta t$, $\Delta u$) increase. In this sense, control effort is the minimal information required to navigate the particles to the target within the target corridor limits with the constraint of a desired arrival probability.

## 3. Model of microswimmer motion and control

The model represents actively self-propelled microswimmers with the possibility to turn the propulsion mechanism on and off. We used models representing microswimmers which achieve propulsion by self-diffusiophoresis [15,37], but the principle remains identical also for other switchable propulsion mechanisms [38–41]. The simulation thus represents partly coated particles that under propulsion experience a force in a fixed direction referring on the particle's shape and are redirected by diffusion, neglecting hydrodynamic interactions.

To represent such experiments in a computer simulation, we rely on previously published models for Janus [15,42] and $L$-shaped microswimmers [21]. We here summarize the model but refer the reader to the original publications for more detail. We restrict the motion to a two-dimensional plane, which may be tilted by an angle $\alpha$ to allow for the influence of gravitation. For all particles, we allow for two translational degrees of freedom in the plane of movement, $\mathbf{r} = (x, y)$, and one rotational degree of freedom, representing the orientation $\varphi$ in that plane, with the orientation vector (figure 1)

$$\mathbf{p} = \begin{pmatrix} -\sin(\varphi) \\ \cos(\varphi) \end{pmatrix}. \tag{3.1}$$

The particles' motion is described as a superposition of active propulsion and stochastic Brownian motion described as a time discrete evaluation of an independent Wiener process calculated with pseudorandom variables $\zeta$. All following equations are defined in the centre of mobility of the particles. The equations of motion for the Janus particles are

$$\dot{\mathbf{r}}_J = v_{\text{act}} \mathbf{p} + \boldsymbol{\zeta_r} \sqrt{2D_{T,J}/\tau} \tag{3.2}$$

and

$$\dot{\varphi}_J = \zeta_\varphi \sqrt{2D_{R,J}/\tau}, \tag{3.3}$$

royalsociety publishing.org/journal/rsos    R. Soc. Open Sci. **8**: 201839

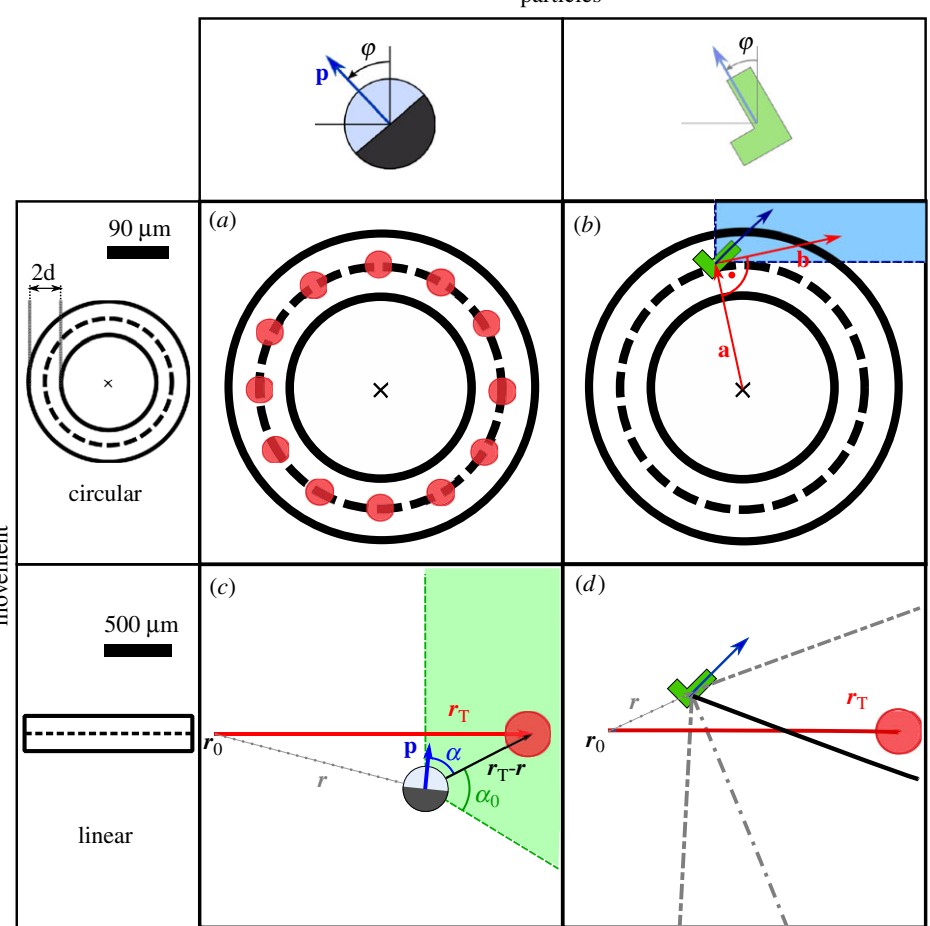

**Figure 1.** Different shapes of particles and movements require different control strategies. *L*-shaped as well as Janus particles are navigated through circular and linear target corridors, where circular corridors have a radius of $r = r_L = 94$ μm and linear corridors a length of $s = 500$ μm. All corridors have a width of $2d = 40$ μm. The particles start at the initial position $r_0$ and are navigated towards the target $r_t$, with $r_0 = r_t$ for circular navigation. (*a,c*) Due to their symmetry in shape, the navigation of Janus particles through a circular corridor requires the definition of sub-targets. Particles are navigated from one target to the other with the same OnOff strategy as for linear navigation. The propulsion is set to $F = F_{max}$ if the particle orientation **p** points towards the shaded area. (*b*) To navigate *L*-shaped particles clockwise in a circle the vector **a** between the centre of the desired circle and the current particle position is defined. The propulsion velocity is chosen based on the particle position within the corridor and its orientation. (*d*) *L*-shaped particles show a very defined and stable set of trajectories under the influence of gravity, depending on the propulsion force. The force level is selected according to the angle $\gamma$ between the vector from the particle position to the target and the *y*-axis. The number of force levels can be chosen (here: $N_F = 4$). Then, the propulsion force is set to the force leading to a trajectory passing nearest to the target (solid black line). For detailed information on the control strategies, see §3.3.

where the first term of equation (3.2) represents the active propulsion with velocity magnitude $v_{act}$ and direction orientation vector **p**, which only influences the translation of the microswimmer. The second term of equations (3.2) and (3.3) represent the Brownian motion with pseudorandom variables $\zeta$ with zero mean and standard deviation of 1 scaled by the translational and rotational diffusion constants of the Janus particles $D_{T,J}$ and $D_{R,J}$, respectively.

Due to the asymmetrical shape, the equations of motion for the Janus and *L*-shaped particles differ, but follow the same logic [21]

$$\dot{\mathbf{r}}_L = \beta F(\underline{\mathbf{D}}_{T,L}\mathbf{u}_\perp + l\mathbf{D}_C) + \zeta_\mathbf{r} + \beta\underline{\mathbf{D}}_{T,L}\mathbf{F}_G \tag{3.4}$$

and

$$\dot{\varphi}_L = \beta F(lD_{R,L} + \mathbf{D}_C\mathbf{u}_\perp) + \zeta_\varphi + \beta\mathbf{D}_C\mathbf{F}_G. \tag{3.5}$$

The first term represents active propulsion by the force *F*, the second term Brownian motion with pseudorandom variables with zero mean and variances $\langle\zeta_\mathbf{r}(t_1) \otimes \zeta_\mathbf{r}(t_2)\rangle = 2\underline{\mathbf{D}}_{T,L}\delta(t_1 - t_2)$, $\langle\zeta_\mathbf{r}(t_1)\zeta_\varphi(t_2)\rangle =$

**Table 1.** Diffusion coefficients $D$ and properties of Janus- and L-particles. The Janus particles' movement can be described by means of translational ($D_T$) and rotational ($D_R$) diffusion coefficients. The movement of L-particles must be described with translational ($D_\parallel$, $D_\perp$, $D_\perp^\parallel$), rotational ($D_R$) and translational-rotational coupling coefficients ($D_C$). Also the geometrical quantities of the microswimmers are given: diameter of the Janus particles ($\sigma$), long ($a$) and short ($b$) L-shaped particle arm, as well as the effective lever arm $l$, from the particle's centre of mass.

| | | |
|---|---|---|
| L-shaped particle | $D_{\parallel,L}$ | $7.2 \times 10^{-3}\ \mu m^2\ s^{-1}$ |
| $l = -0.75\ \mu m$ | $D_{\perp,L}$ | $8.1 \times 10^{-3}\ \mu m^2\ s^{-1}$ |
| $\beta = 1/(k_B T)$, $T = 305$ K | $D_{\perp,L}^\parallel$ | $0\ \mu m^2\ s^{-1}$ |
| $a = 9\ \mu m$ | $D_{C,L}^\parallel$ | $5.7 \times 10^{-4}\ \mu m\ s^{-1}$ |
| $b = 6\ \mu m$ | $D_{C,L}^\perp$ | $3.8 \times 10^{-4}\ \mu m\ s^{-1}$ |
| | $D_{R,L}$ | $6.2 \times 10^{-4}\ s^{-1}$ |
| Janus particle | $D_{T,j}$ | $2.7 \times 10^{-2}\ \mu m^2\ s^{-1}$ |
| $\sigma = 4.2\ \mu m$ | $D_{R,j}$ | $8.3 \times 10^{-3}\ s^{-1}$ |
| Janus particle | $D_{T,J}$ | $0.01\ \mu m^2\ s^{-1}$ |
| $\sigma = 9.96\ \mu m$ | $D_{R,J}$ | $6.2 \times 10^{-4}\ s^{-1}$ |

$2\mathbf{D}_C \delta(t_1 - t_2)$ and $\langle \zeta_\varphi(t_1)\zeta_\varphi(t_2)\rangle = 2\,D_{R,L}\delta(t_1 - t_2)$ [21]. The last term is an additional term which considers gravitational forces if the plane is tilted ($\alpha \neq 0$) [21]. The model parameters are described and listed in table 1.

## 3.1. Numerical simulations

The simulated stochastic motion depends on the diffusion constants $D$ for Janus and L-shaped particles, respectively (table 1). For the L-shaped particles with the given dimensions (table 1), the diffusion coefficients were obtained experimentally from short-time correlation experiments without gravity and passive sedimentation experiments [21]. The diffusion coefficients for Janus particles with diameter $\sigma = 4.2\ \mu m$ have also been experimentally determined [15]. For comparison with theory these diffusion coefficients had also been calculated by solving the Stokes equation [20] and good agreement with the experimental values has been found [21]. The simulations of motion for both particles were based on a time discrete evaluation of the equations of motion for the two translational degrees of freedom, $x$ and $y$, and the rotational degree of freedom $\varphi$. The differential equations for the particles' positions (equations (3.2) and (3.4)) and orientations (equations (3.3) and (3.5)) were solved for constant time intervals of $\tau = 0.5$ s

$$\mathbf{r}_{i+1} = \mathbf{r}_i + \dot{\mathbf{r}}_{J,L} \cdot \tau \qquad (3.6)$$

and

$$\phi_{i+1} = \phi_i + \dot{\varphi}_{J,L} \cdot \tau. \qquad (3.7)$$

## 3.2. Comparability of different particle shapes

To allow for a comparison of the two different particle shapes, two parameters are crucial: the propulsion velocity or equivalently the propulsion force and the rotational diffusion of the particle, as the particle will be actively driven, whenever it points in the right direction towards the goal. This change of direction is completely due to the rotational diffusion of the particle and is not effected by any control strategy.

Active propulsion of the particles is parameterized differently for both morphologies. For the Janus particles, it is parameterized via the velocity $v_{act}$ (equations (3.2) and(3.3)), while in the L-particles via the propulsion force $F$ (equations (3.4) and(3.5)). As we set a propulsion velocity for the Janus particles and a propulsion force for the L-shaped particles in our simulations, we decided on a maximum velocity for both particles and adjusted the propulsion force of the L-shaped particles, accordingly. To obtain

comparable results for both particle shapes, the maximal propulsion velocity without the influence of gravity was set to $v_{act} = 2.83\,\mu\text{m s}^{-1}$. This velocity could be used as direct input for the Janus simulation. For the *L*-shaped particles, the same maximum velocity is given at the propulsion force of $F_{max} = 1.47\,\mu\text{N}$.

However, comparing the diffusion coefficients for *L*-shaped particles with $l = -0.75\,\mu\text{m}$ and Janus particles with diameter $\sigma = 4.2\,\mu\text{m}$, it is evident that the rotational as well as the translational diffusion coefficients of the Janus particles are at least one magnitude higher than those of the *L*-shaped particles. Thus, the Janus particles are faster and much more agile. To make a fair comparison between both particle morphologies, we introduce larger Janus particles for which we calculated the diameter such that the rotational diffusion coefficients of *L*-shaped and Janus particles match. Solving the equation for the diffusion of Janus particles

$$D_{R,J} = \frac{k_B T}{8\pi\eta\sigma^3},$$

(3.8)

for the diameter $\sigma$ and using the experimentally determined parameters for the small Janus particles ($\sigma_j = 4.2\,\mu\text{m}$, $T = 305\,\text{K}$, $\eta_{\text{water}-2.6\text{lutidine}} = 0.0022\,\text{kg m}^{-1}\,\text{s}^{-1}$), that also apply for another size of spherical particle, we can calculate the diameter for a larger Janus particle with a diffusion parameter comparable to those of the used *L*-shaped particles

$$\sigma_J = 9.96\,\mu\text{m}.$$

(3.9)

From here on, we term the small Janus particles *j*-particles and the large Janus particles *J*-particles and also use upper and lower case as indices, accordingly.

## 3.3. Control strategies

As stated in the hypothesis, we investigated circular and linear movement. To achieve such movements, we require active control strategies to overcome the inherent stochastic motion. Roughly speaking, only if the particles orientation is aligned with the movement goal is active propulsion turned on. Thus, every control strategy navigates the particles to the target position by setting the propulsion force and thereby the propulsion velocity dependent upon the current particle position $\mathbf{r} = (x, y)$ and orientation $\mathbf{p}$. Hereby, the stochastic reorientation of the particles is exploited by the control strategies [11,12,15].

All movements are defined by an initial position $\mathbf{r}_0 = (0, 0)$ and a target position $\mathbf{r}_T = (x_T, y_T)$. The movement is completed correctly if the particle reaches the target without leaving a pre-defined target corridor. The width of the corridor is set to $d = 20\,\mu\text{m}$ orthogonal in both directions from the direct path between start and goal, resulting in a total width of $d = 40\,\mu\text{m}$ of the corridor. The particle is placed in the middle initially (figure 1). If a centre of mass of the particle leaves the corridor, it is counted as a fail. In every condition, we simulated 500 runs and determined the arrival probability $p$. The control strategies have free parameters (see below), which were optimized for the arrival probability.

One remark in advance: the different morphologies of the particles require different control strategies to navigate them to the target. For example, *L*-shaped particles in a tilted setup generate different trajectories for different propulsion forces [21]. This can be exploited by a more complex control strategy which not only switches propulsion on and off, but also controls the propulsion force. A transfer of this control scheme to the spherical Janus particles would bring no benefit as varying propulsion force does not change the large-scale movement (it remains stochastic) and therefore would only reduce the distance travelled towards the target. Also, introducing gravity by the tilted setup only benefits the *L*-shaped particles, while for a Janus particle it will simply add a sedimentation force. The control strategies described below have been explicitly chosen to exploit the particles' respective benefits.

### 3.3.1. Janus, OnOff strategy

The control strategy for the Janus particles has been introduced before [15]. Here, propulsion depends on the angle $\alpha$ between the particle orientation $\mathbf{p}(t)$ and the vector connecting the target position $\mathbf{r}_T$ and the

current particle position $\mathbf{r}(t)$ (figure 1c):

$$\alpha = \arccos\left(\frac{\mathbf{p} \cdot (\mathbf{r}_T - \mathbf{r})}{|\mathbf{p}||(\mathbf{r}_T - \mathbf{r})|}\right). \tag{3.10}$$

When $\alpha$ is smaller than or equal to a given threshold angle $\alpha_0$, the propulsion velocity is set to $v = v_{\max}$ and $v = 0$ otherwise, i.e.

$$v_{\mathrm{OnOff}} = \begin{cases} v_{\max} & \alpha \leq \alpha_0 \\ 0 & \alpha > \alpha_0. \end{cases} \tag{3.11}$$

This control strategy is valid for both linear and circular movements and the parameter $\alpha_0$ is optimized. For circular movements, the targets are placed along the circle (figure 1a).

### 3.3.2. Ls, circular strategy

Anti-$L$-shaped particles will perform a clockwise rotation when a force is applied at the center of the short arm (see §3). To navigate $L$-shaped particles in a circle, two vectors are defined (figure 1b): the vector $\mathbf{a} = \mathbf{r}_c - \mathbf{r}$ is the connecting vector between the centre of the desired circle $\mathbf{r}_c$ and the current particle position $\mathbf{r}$. The direction of the long swimmer arm is given by $u_\perp$ (3.4) and (3.5). Again, the particle is meant to reach the target position without leaving the predefined target corridor. Depending on the particle position within the target corridor different threshold angles are applied for choosing the propulsion velocity. The force is set to $F = F_{\max}$ in one of the following conditions:

$$(1) \qquad |\mathbf{a}| < R_L + \frac{\mathrm{d}r}{2} \quad \& \quad |\mathbf{a}| > R_L - \frac{\mathrm{d}r}{2}, \tag{3.12}$$

$$\angle(u_\perp, \mathbf{a}) \leq \alpha_{\mathrm{limit},1} \quad \& \quad \angle(u_\perp, \mathbf{a}) \leq 90° \tag{3.13}$$

$$(2) \qquad |\mathbf{a}| > R_L + \frac{\mathrm{d}r}{2} \quad \& \quad |\mathbf{a}| < R_L + dr, \tag{3.14}$$

$$\angle(u_\perp, \mathbf{a}) \geq \alpha_{\mathrm{limit},2} \quad \& \quad \angle(u_\perp, \mathbf{a}) \leq 90°. \tag{3.15}$$

The threshold angles $\alpha_{\mathrm{limit}}$ have to be optimized for the radius $R_L$ of the desired circle of the movement.

### 3.3.3. Ls, linear strategy

On a level plane, no valid control strategy for $L$-particles was found. Here, the natural reorientation results in circular movements leaving the target corridor. However, under the influence of gravity ($\alpha = 10.67°$), $L$-shaped particles show a very defined and stable set of trajectories which depend on the propulsion force (see also [21]). They sediment downwards due to gravity without active propulsion. With active propulsion, they always yield a movement directed in positive $x$-axis with different angles relative to the $x$-axis. The range of possible angles is limited by the sedimentation trajectory and the highest trajectory through active propulsion (figure 1d).

For the linear strategy, the applied force is divided into $N_F$ equal steps within a range $0 < F < F_{\max}$. The force level is selected according to the angle $\gamma$, which is measured between the vector from the particles' position to the target and the $y$-axis: $\gamma = \angle(r_{l \to \mathrm{target}}, \hat{e}_y)$ (figure 1d). This way, $L$-particles can be navigated to a target on the $x$-axis within the linear target corridor.

## 4. Results

### 4.1. Circular movement

Without Brownian motion, self propelled $L$-shaped particles move in a circle with radius $r_L = 94\ \mu\mathrm{m}$. The time required for this movement is $T = 618\ \mathrm{s}$. In this scenario, naturally all $L$-shaped particles arrive at the target line resulting in an arrival probability of $p_{L,\mathrm{nonoise}} = 100\%$. Adding Brownian noise without any control results in an arrival probability of $p_{L,\mathrm{nocontrol}} = 0\%$, as all particles are randomly reoriented by rotational diffusion and therefore eventually exit the allowed target corridor. With the $\mathrm{OnOff}_L$ control strategy, the arrival probability rises to $p_L(\Delta t = 2\ \mathrm{s}) = 77\%$.

Self-propelled Janus particles move in a straight line without Brownian motion and show a characteristic random behaviour when considering Brownian motion. In both cases, the arrival

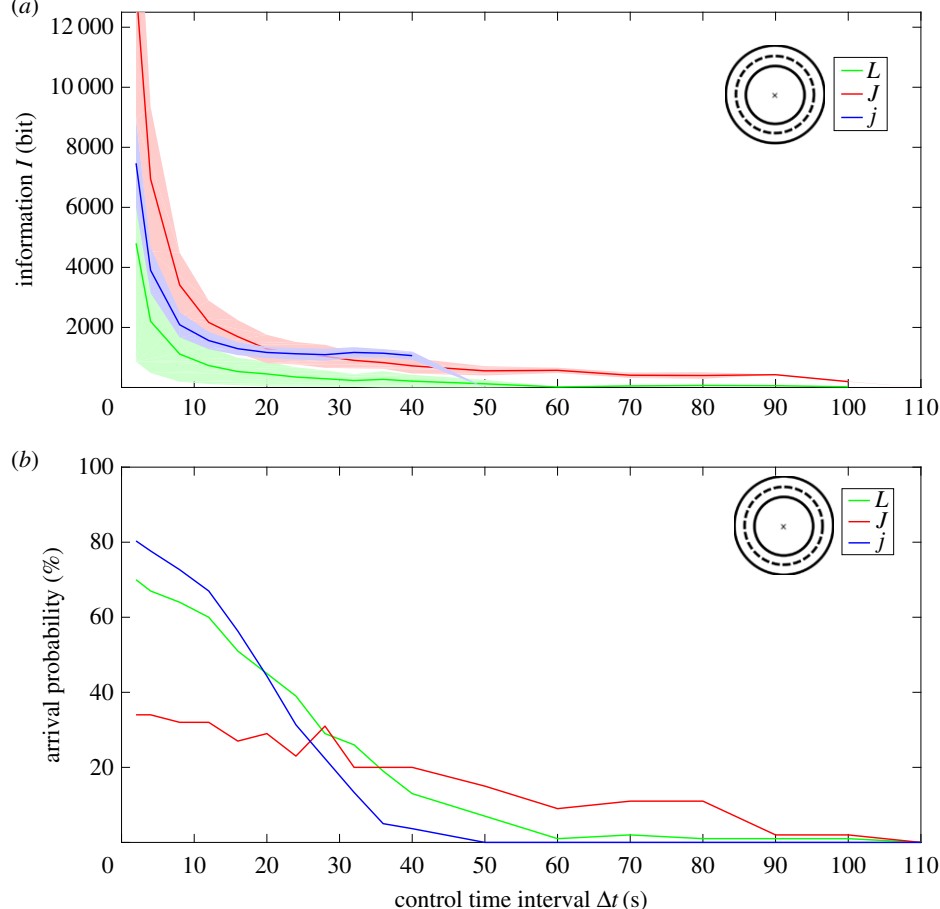

**Figure 2.** Control effort and arrival probability in circular movement depend on the particle's morphology. All particles had to complete a full circle with radius $r = r_0$—which is the natural radius of the $L$-particles—and identical corridor width ($d = 20\ \mu m$, see (figure 1)). (a) The processed information represents the control effort and varies with time interval $\Delta t$. It is always smaller for $L$-shaped particles' (green line) than for small (blue) and large (red) Janus particles. (b) The arrival probability also varies with the time interval $\Delta t$. For small time intervals $\Delta t$, the arrival probability for small Janus particles is highest ($p_j(\Delta t = 2\ s) = 80\%$). With larger time intervals, the arrival probabilities for all particles decrease. For large time intervals, the arrival probability is highest for the large Janus particles ($p_j(\Delta t = 80\ s) = 18\%$). This shows that control effort of $L$-shaped particles is lower than with Janus particles for circular movement, but this may come at the cost of arrival probability.

probability is $p_{J,\text{nonoise}} = 0\%$, $p_{J,\text{nocontrol}} = 0\%$, and $p_{j,\text{nonoise}} = 0\%$, $p_{j,\text{nocontrol}} = 0\%$ for small and large Janus particles. With the control strategy $\text{OnOff}_{J,j}$, the arrival probability rises to $p_J(\Delta t = 2\ s) = 34\%$ and $p_j(\Delta t = 2\ s) = 80\%$, respectively (figure 3a).

The information processed in these cases is determined with equation (2.1). The logarithm in equation (2.1) is always equal to one, as all OnOff strategies only switch between $u^{\min} = 0$ and $u^{\max} = v_0$ with $\Delta u = v_0$. Only the time resolution $\Delta t$ and the duration of the movement $T_t$ determine the information $I$. Here, we set the controller time resolution to the simulation update time $\Delta t = \tau = 2\ s$. The processed information varies between particles. It is lowest for the $L$ particles with $I_L(\Delta t = 2\ s) = 4803$ bit and $I_j(\Delta t = 2\ s) = 13\ 320$ bit for the large and $I_j(\Delta t = 2\ s) = 7469$ bit for the small Janus particles. This reflects only the duration $T_t$ of the movement, as all other values in equation (2.1) are identical.

However, the processed information depends on the control time interval $\Delta t$ (equation (2.1)). Increasing $\Delta t$ reduces the information $I$ for all three particles (figure 2a). The information of the $L$ particles $I_L$ is always lower than $I_J$ and $I_j$ for $2\ s \leq \Delta t \leq 110\ s$. However, the arrival probability also decreases with increasing time resolution (figure 2b).

Control effort is then defined as the minimum information required to generate a movement with specific constraints. If one requires an arrival probability of at least $p = 40\%$, large $J$ particles are not adequate while $L$'s and $j$'s achieve $p = 40\%$ and more for $\Delta t = 20\ s$. However, $L$'s require only $I_L(\Delta t =$

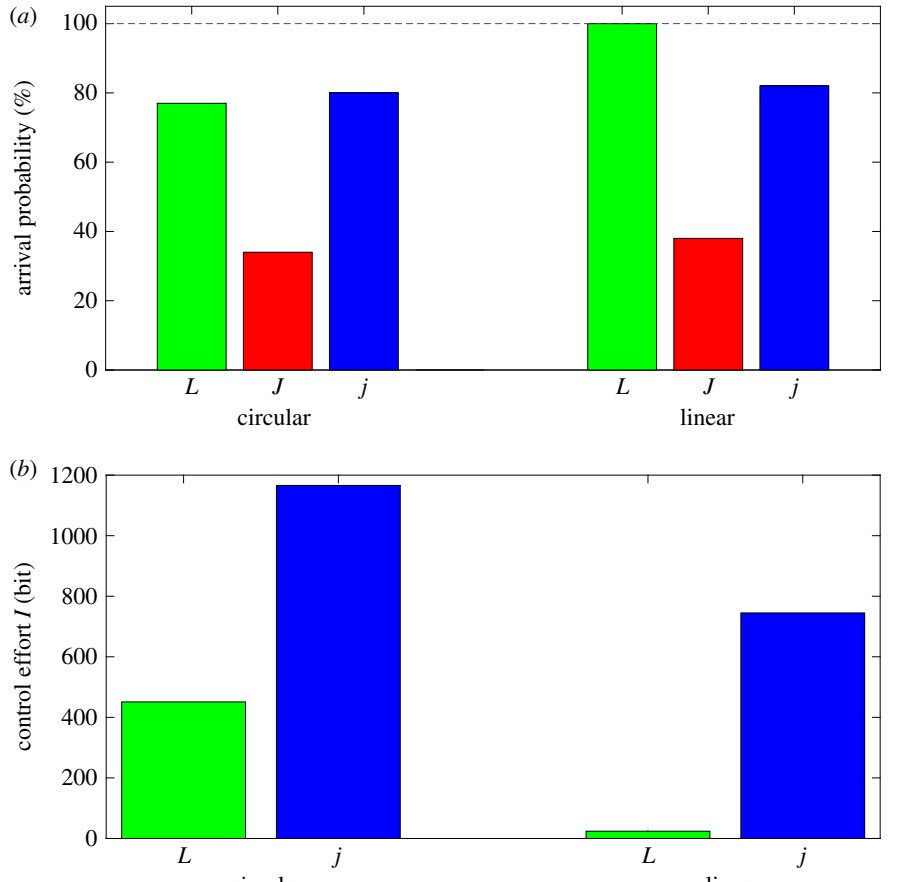

**Figure 3.** (a) Arrival probability for all particles for circular ($r = r_L$) and linear ($s = 500\ \mu m$) navigation with the respective control strategies. (b) Control effort calculated from equation (2.1) for circular and linear navigation. As only L-shaped and j-particles reach arrival probabilities $p > 40\%$, J-particles are not depicted here.

32 s) = 451 bit when compared with $I_j(\Delta t = 20\ s) = 1166$ bit and, therefore, the control effort is more than two times lower for $L$ particles (figure 3b).

Other limitations may prefer other particles. If the limitation is a time resolution of $\Delta t = 60$ s, the arrival probability of J's would be highest ($p_J(\Delta t = 60\ s) = 9\%$).

For the simulations above, the target radius was the natural radius of the $L$-particles ($r = R_L$). For other radii, the arrival probability of the $L$-particles decreases and the information increases (figure 4). For rather large radii with $r > 4R_L$, no $L$-particles arrive. This is very similar also for $J$ and $j$ particles (see electronic supplementary material, figure S1). However, Janus particles show no minimum of information for a specific radius—the required information increases with decreasing radii (figure 5).

## 4.2. Linear movement

Navigating the particles towards a goal through a linear target corridor is also possible with all three particles. For small distances all particles arrive, independent of their shape. For increasing distances, the arrival probability for Janus particles starts to decrease first. At a relatively large distance of $d = 500\ \mu m$ and small control time interval ($\Delta t = 2$ s), all $L$-particles still arrive, while only 38% of $J$-particles and 80% of $j$-particles reach the target. The information required to control the particles is low for the $L$-particles and higher for the Janus-particles (figures 6 and 3a).

The control strategy of the $L$-shaped particles allows for a continuous variation of the propulsion force signal $u$. To calculate the information, the signal is discretized by $\Delta u = 1/N_F$, where $N_F$ specifies the number of allowed force levels. Choosing very few force levels ($N_F = 4$) results in low information and, for a small control time interval of $\Delta t = 2$ s, in a high arrival probability. However,

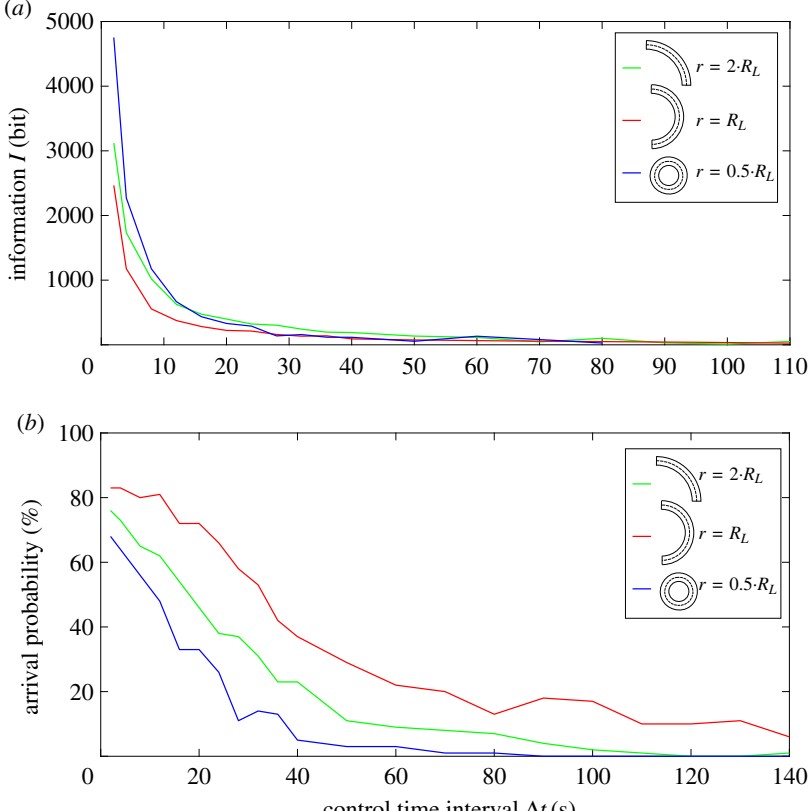

**Figure 4.** Control effort is lowest for L-shaped particles, if the target circle radius $r$ corresponds to the particles' natural radius $r = R_L$ (red line in subfigure $a$). In this case, the arrival probability is also highest (red line in subfigure $b$). Increasing or decreasing the target circle radius $r$ always increases control effort and reduces arrival probability. A fair comparison is achieved by choosing only a partial segment of the circle such that for each radius the total travel distance between start and target line along the circle was identical in all three settings. This shows that the natural radius can be exploited for control effort and arrival probability in circular movement.

increasing the time interval results in rapidly decreasing arrival probabilities, also for the Janus particles (figure 6).

It is further possible to improve the arrival probability even for large $\Delta t$ by increasing the number of possible force levels $N_F$. E.g. if one requires an arrival probability of at least $p = 40\%$, as above, the control effort in the L-particles can be as low as $I_L(\Delta t = 800\,\text{s}, \Delta u = 1/6) = 24$ bit, while $j$ particles require at least $I_j(\Delta t = 28\,\text{s}) = 745$ bit and J particles do not reach $p = 40\%$ even for small control time intervals $\Delta t$ (figure 3$b$).

# 5. Discussion

The core idea behind the measure *control effort* is to quantify the contribution of the morphology to the generation of a movement [7,35]. This work showed that our information entropy-based measure can be applied to actively controlled microscopic systems. Actively controlling a movement and steering a particle towards a target requires processing information about the system's state for a timestep based repetitive decision on the appropriate control command by means of a control strategy. Although we use the approach to reduce the information available for the control strategy to measure control effort, finding this minimal information for the actual operation of a controller is not the goal. The discretization is rather used to quantify the contribution of the morphology for a given task. Previous work had already shown that the morphology of the microswimmer crucially effects the stochastic as well as the deterministic movement [20,21]. Thus the morphology can be beneficial—but also disadvantageous—to the accomplishment of a specific task, which becomes evident in a large or small value of control effort.

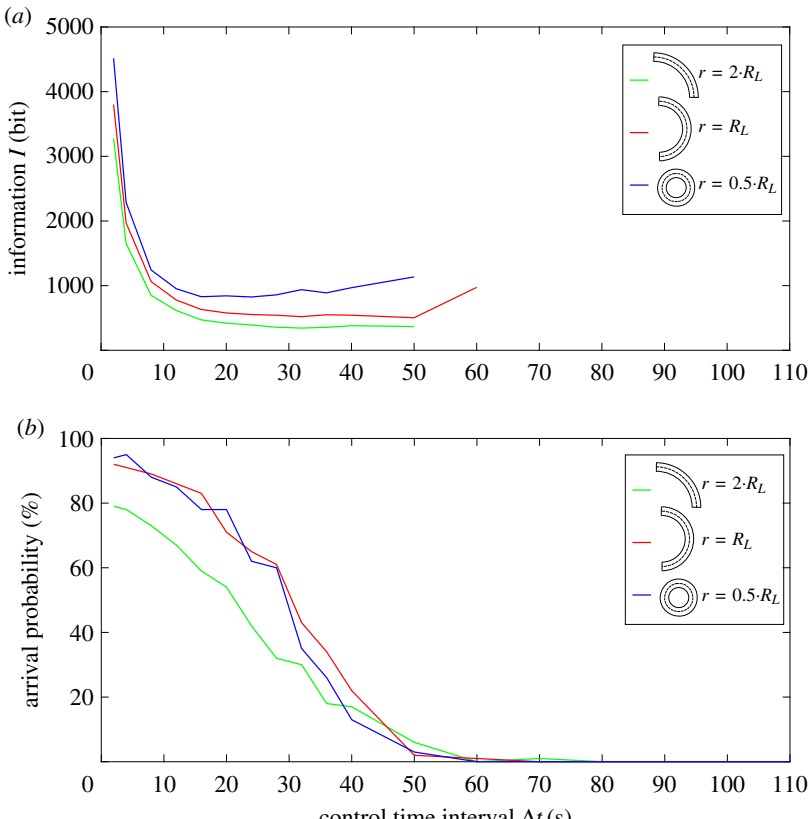

**Figure 5.** Varying the radius of the target circle $r$ in Janus particles (here: $j$) also influences processed information and arrival probability. In contrast to $L$-shaped particles, the processed information is lowest for the smallest curvature ($r = 2R_L$, green line subfigure $a$), as is the arrival probability (green line subfigure $b$). This means that Janus particles do not show an optimal target radius where control effort would be minimal.

For this study, we chose two morphologies of microswimmers which had already been modelled and studied in experiments [9,20,21,37]. The results confirmed expectation (1) we proposed in the introduction: the natural rotation of self-propelled $L$-shaped particles could be exploited for circular movements reducing their control effort when compared with symmetric Janus particles. For linear movements, the control effort is smaller for Janus than $L$-shaped particles if both swim in a level setup. However, by slightly tilting the setup and therefore introducing a gravitational component, the emerging rich trajectory repertoire of the $L$-shaped particles [21] can be exploited to design a controller which requires even less control effort than the level Janus particles for linear movements. This confirms expectation (2) from the introduction. So the choice of microswimmers with different morphologies and thereby movement characteristics can be significant in supporting the movement. However, it will always strongly depend on the type of task and its environment. This means that control effort cannot generally decide whether one particle is better than the other, but can only evaluate the contribution of the morphology in the context of the task and the environment. For completeness, it is also important to note that the morphology of the $L$-shaped particles limits the possible targets as no controller can be designed to steer them in negative $x$-direction in the tilted setup or to change their circular movement direction on the plane. This means that there is a trade-off between *control generality* and *control effort*.

The basis for our models were active synthesized particles with a photophoretic self-propulsion mechanism [9,20,37]. However, the control approach (on–off strategy) and the evaluation (control effort) would also be applicable to systems with other propulsion mechanisms which allow online on/off switching of the active propulsion [38–41]. It may even be feasible to design a morphology (and diffusion matrix) to optimize for a specific movement goal. Currently, our approach to measure control effort relies on reducing the information content by changing the discretization intervals. This limits the direct applicability to other, biologically more plausible and relevant control strategies, e.g. run-and-tumble in chemotactic species [25,43,44] which rely on a continuous sampling of a chemical

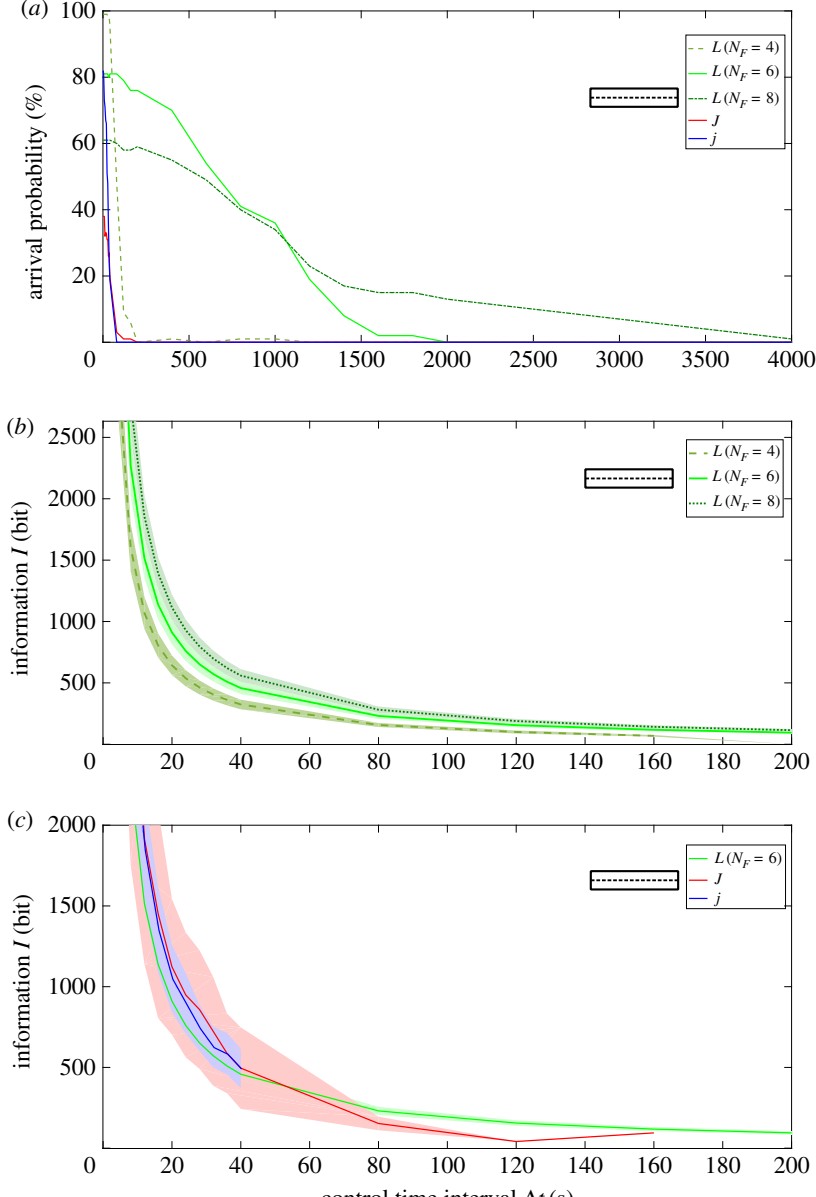

**Figure 6.** Linear movement: particles are navigated to a target at a linear distance of $d = 500\ \mu m$ within a linear target corridor. For linear movement, $L$-particles show the highest arrival probability, independent of the force resolution of their control strategy ($N_F = 4, 6, 8$) (green lines, subfigure $a$). Allowing more values increases the control information as more values are processed (subfigure $c$, also see electronic supplementary material, figure S5). However, the comparison of information is made between Janus particles and $L$-particles with the highest arrival probability ($N_F = 6$) (subfigure $c$, lines end, when arrival probability of the particle equals zero). Taking into account arrival probability and information, $L$-particles have a much smaller control effort than Janus particles (figure 3) This shows that for linear movement, the morphology of $L$-shaped particles can also be exploited to reduce control effort.

concentration. However, limiting the information content could also be achieved by other approaches, e.g. for continuous Gausian variables [45]. We speculate that such an extended approach could also be applied to models of biological microswimmers and may reveal potentially information efficient behaviour as one optimization criterion for the biological control strategy.

We believe that this work demonstrates that quantifying the contribution of morphology to the generation of movement is not only relevant to understand biomechanics of macroscopic animals [35] but also an interesting measure for microswimmers and hope to inspire other researchers to take this into account as a possible measure or even figure of merit in the control and design of microswimmers.

Data accessibility. All data was created using MATLAB scripts (with MATLAB version 2016a). The scripts can be accessed via the following public link: https://doi.org/10.18419/darus-1179.

Authors' contributions. J.M.R. realized the simulations and analysis of the same and helped draft the manuscript. C.H. revised the study critically, especially for the microswimmers content. S.S. revised the study critically, especially for the control effort content. D.F.B.H. conceived of the study, participated in data analysis and helped draft the manuscript. All authors gave final approval for publication.

Competing interests. We declare we have no competing interests.

Funding. The research of DH was supported by the Ministry of Science, Research and the Arts Baden-Württemberg (Az: 33-7533.-30-20/7/2). J.R. and C.H. also thank the Deutsche Forschungsgemeinschaft (DFG) for funding the research through the SPP 1726 'Microswimmers: from single particle motion to collective behaviour' (grant no. HO1108/24-2). C.H. and S.S. furthermore acknowledge support through the DFG – Project Number 390740016 – EXC 2075 (SimTech).

Acknowledgements. We thank Clemens Bechinger for the discussions that led to this study and the his valuable information on the experimental counterpart of the simulated particles.

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
