## [Peer Review File · Royal Society Open Science]

Review History

RSOS-201839.R0 (Original submission)

Review form: Reviewer 1

Is the manuscript scientifically sound in its present form?

No

Are the interpretations and conclusions justified by the results?

No

Is the language acceptable?

Yes

Do you have any ethical concerns with this paper?

No

Have you any concerns about statistical analyses in this paper?

No

Recommendation?

Reject

Comments to the Author(s)

Please read the PDF (see Appendix A).

Review form: Reviewer 2

Is the manuscript scientifically sound in its present form?

Yes

Are the interpretations and conclusions justified by the results?

Yes

Is the language acceptable?

Yes

Do you have any ethical concerns with this paper?

No

Have you any concerns about statistical analyses in this paper?

No

Recommendation?

Accept with minor revision (please list in comments)

Comments to the Author(s)

The manuscript contains a new approach to quantify the information effort needed to steer a microrobot or an artificial microswimmer to follow a predefined path. The research builds on previous works of the authors and generalizes it to a non-spherical particle.

The generalization might prove useful for the design of general-shaped microswimmers. However, there are a few things which I would like the authors to address before the manuscript is ready for publication.

- The control strategies are discussed for a specific model of particles which have anisotropic diffusivity. At the same time, the authors neglect hydrodynamic interactions with walls. Could the authors comment on how much these interactions could modify the presented framework. If a particle was moving close to a wall which would (a) slow down the particle and (b) introduce additional anisotropy of the motion of the particle, e.g. an additional reorientation. These factors would influence the analysed motion in a channel and should be either accounted for or discussed. I fully agree that the presented approach is valid far from boundaries when the motion is affected solely by the externally controlled propulsion which couples to the hydrodynamic anisotropy of the particle but in the presence of nearby walls this description would lose its validity. Perhaps it would be better in this context not to call these predefined trajectories 'channels' but somehow to reflect the fact that there are no physical boundaries which would interact with the swimmer hydrodynamically?

- I was wondering whether this treatment can be generalised to an arbitrary diffusion matrix and thus to a general-shaped particle. Since the control effort is highly specific for the geometry of the particle, the development of a general framework could be an interesting way forward and I was wondering if the authors could consider a general-shaped particle?

- There is a number of systems in which external factors can switch on/off active propulsion.

Examples are:

Feng et al Appl Phys Rev (2021)

Vutukuri et al Nature Comm (2020)

Dong et al ACS Nano (2016)

Jiang et al Phys Rev Lett (2010)

and others in literature. It would be beneficial if the authors could discuss the applicability of their findings to such systems.

- Could the Authors comment more on whether L-shaped particles with the ability to switch their propulsion on/off have been synthesized?

A couple of minor points:

- the Authors should use uniform notation for the decimal point (not comma) throughout the paper.

- in the introduction, when listing papers involving control strategies, the Authors could specifically comment on what these strategies are based on.

- In Table 1 and above eq. (6), the parameter ℓ seems not to be defined (or at least I could not find the definition. It would be helpful if the authors could clarify this point

- the symbol T is used both for temperature and time required for movement. Also, T is not defined in Eq. (14).

- The last sentence of Sec. 2 is not clear.

Decision letter (RSOS-201839.R0)

Dear Ms Riede

The Editors assigned to your paper RSOS-201839 "The control effort to steer self-propelled microswimmers depends on their morphology: comparing symmetric spherical vs. asymmetric L-shaped particles" have now received comments from reviewers and would like you to revise the paper in accordance with the reviewer comments and any comments from the Editors. Please note this decision does not guarantee eventual acceptance.

Please submit your revised manuscript and required files (see below) no later than 21 days from today's (ie 01-Jun-2021) date. Note: the ScholarOne system will 'lock' if submission of the revision is attempted 21 or more days after the deadline. If you do not think you will be able to meet this deadline please contact the editorial office immediately.

on behalf of Dr Kenta Ishimoto (Associate Editor) and Pietro Cicuta (Subject Editor)
openscience@royalsociety.org

Associate Editor Comments to Author (Dr Kenta Ishimoto):

The manuscript has been reviewed by the two referees, and numerous concerns are raised. I would like to ask the authors to properly address all the comments raised by the referees.

Reviewer comments to Author:

Reviewer: 1

Comments to the Author(s)

Please read the attached PDF

Reviewer: 2

Comments to the Author(s)

The manuscript contains a new approach to quantify the information effort needed to steer a microrobot or an artificial microswimmer to follow a predefined path. The research builds on previous works of the authors and generalizes it to a non-spherical particle.

The generalization might prove useful for the design of general-shaped microswimmers.

However, there are a few things which I would like the authors to address before the manuscript is ready for publication.

- The control strategies are discussed for a specific model of particles which have anisotropic diffusivity. At the same time, the authors neglect hydrodynamic interactions with walls. Could the authors comment on how much these interactions could modify the presented framework. If a particle was moving close to a wall which would (a) slow down the particle and (b) introduce additional anisotropy of the motion of the particle, e.g. an additional reorientation. These factors would influence the analysed motion in a channel and should be either accounted for or discussed. I fully agree that the presented approach is valid far from boundaries when the motion is affected solely by the externally controlled propulsion which couples to the hydrodynamic

anisotropy of the particle but in the presence of nearby walls this description would lose its validity. Perhaps it would be better in this context not to call these predefined trajectories 'channels' but somehow to reflect the fact that there are no physical boundaries which would interact with the swimmer hydrodynamically?

- I was wondering whether this treatment can be generalised to an arbitrary diffusion matrix and thus to a general-shaped particle. Since the control effort is highly specific for the geometry of the particle, the development of a general framework could be an interesting way forward and I was wondering if the authors could consider a general-shaped particle?

- There is a number of systems in which external factors can switch on/off active propulsion. Examples are:

Feng et al Appl Phys Rev (2021)

Vutukuri et al Nature Comm (2020)

Dong et al ACS Nano (2016)

Jiang et al Phys Rev Lett (2010)

and others in literature. It would be beneficial if the authors could discuss the applicability of their findings to such systems.

- Could the Authors comment more on whether L-shaped particles with the ability to switch their propulsion on/off have been synthesized?

A couple of minor points:

- the Authors should use uniform notation for the decimal point (not comma) throughout the paper.

- in the introduction, when listing papers involving control strategies, the Authors could specifically comment on what these strategies are based on.

- In Table 1 and above eq. (6), the parameter ℓ seems not to be defined (or at least I could not find the definition. It would be helpful if the authors could clarify this point

- the symbol T is used both for temperature and time required for movement. Also, T is not defined in Eq. (14).

- The last sentence of Sec. 2 is not clear.

===PREPARING YOUR MANUSCRIPT===

While not essential, it will speed up the preparation of your manuscript proof if accepted if you format your references/bibliography in Vancouver style (please see

<https://royalsociety.org/journals/authors/author-guidelines/#formatting>). You should include DOIs for as many of the references as possible.

===PREPARING YOUR REVISION IN SCHOLARONE===

Author's Response to Decision Letter for (RSOS-201839.R0)

See Appendix B.

RSOS-201839.R1 (Revision)

Review form: Reviewer 1

Is the manuscript scientifically sound in its present form?

Yes

Are the interpretations and conclusions justified by the results?

Yes

Is the language acceptable?

Yes

Do you have any ethical concerns with this paper?

No

Have you any concerns about statistical analyses in this paper?

No

Recommendation?

Accept as is

Comments to the Author(s)

I wish to thank the authors of 'The control effort to steer self-propelled micro swimmers depends on their morphology: comparing symmetric spherical vs. asymmetric L-shaped particles' for their responses and revisions. They have cleared up my confusion about the work. Furthermore the aims and methods of the manuscript is now much clearer and does reflect their premise. Though I am not 100% sold on the usefulness of 'control effort' to the microscopic swimmer community, I believe the manuscript is now sufficiently sound for publication.

The authors may be interested to note that the importance of geometry on motion and strategy is well known in the microswimming world, as viscous fluid flow is strongly influenced by geometry and microorganisms would not be able to swim without it.

Review form: Reviewer 2

Is the manuscript scientifically sound in its present form?

Yes

Are the interpretations and conclusions justified by the results?

Yes

Is the language acceptable?

Yes

Do you have any ethical concerns with this paper?

Yes

Have you any concerns about statistical analyses in this paper?

No

Recommendation?

Accept as is

Comments to the Author(s)

The authors have answered all my queries from the first report. However, I find the criticism of Reviewer 1 justified and valid. In my opinion, the authors have addressed the points made by Reviewer 1. The result is not entirely satisfying, since the goal proposed by the Authors is neither general nor really useful practically but the manuscript (with these limitations of scope, as aptly described by Reviewer 1), being rather specific, is still an interesting contribution in the sense that it might serve as an inspiration for further works. I would therefore recommend the manuscript for publication.

The work still contains a number of typos, which I am sure the Authors will correct.

Decision letter (RSOS-201839.R1)

Dear Ms Riede

On behalf of the Editors, we are pleased to inform you that your Manuscript RSOS-201839.R1 "The control effort to steer self-propelled microswimmers depends on their morphology: comparing symmetric spherical vs. asymmetric L-shaped particles" has been accepted for publication in Royal Society Open Science subject to minor revision in accordance with the

referees' reports. Please find the referees' comments along with any feedback from the Editors below my signature.

Please submit your revised manuscript and required files (see below) no later than 7 days from today's (ie 24-Aug-2021) date. Note: the ScholarOne system will 'lock' if submission of the revision is attempted 7 or more days after the deadline. If you do not think you will be able to meet this deadline please contact the editorial office immediately.

on behalf of Dr Kenta Ishimoto (Associate Editor) and Pietro Cicuta (Subject Editor)
openscience@royalsociety.org

Associate Editor Comments to Author (Dr Kenta Ishimoto):

Associate Editor: 1

Comments to the Author:

I thank the authors for their efforts on improving the manuscript. As the referee 2 suggests, it is worth proofreading once more before publication.

Reviewer comments to Author:

Reviewer: 1

Comments to the Author(s)

I wish to thank the authors of 'The control effort to steer self-propelled micro swimmers depends on their morphology: comparing symmetric spherical vs. asymmetric L-shaped particles' for their responses and revisions. They have cleared up my confusion about the work. Furthermore the aims and methods of the manuscript is now much clearer and does reflect their premise. Though I am not 100% sold on the usefulness of 'control effort' to the microscopic swimmer community, I believe the manuscript is now sufficiently sound for publication.

The authors may be interested to note that the importance of geometry on motion and strategy is well known in the microswimming world, as viscous fluid flow is strongly influenced by geometry and microorganisms would not be able to swim without it.

Reviewer: 2

Comments to the Author(s)

The authors have answered all my queries from the first report. However, I find the criticism of Reviewer 1 justified and valid. In my opinion, the authors have addressed the points made by Reviewer 1. The result is not entirely satisfying, since the goal proposed by the Authors is neither general nor really useful practically but the manuscript (with these limitations of scope, as aptly described by Reviewer 1), being rather specific, is still an interesting contribution in the sense that it might serve as an inspiration for further works. I would therefore recommend the manuscript for publication.

The work still contains a number of typos, which I am sure the Authors will correct.

===PREPARING YOUR MANUSCRIPT===

===PREPARING YOUR REVISION IN SCHOLARONE===

Author's Response to Decision Letter for (RSOS-201839.R1)

See Appendix C.

Decision letter (RSOS-201839.R2)

Dear Ms Riede,

I am pleased to inform you that your manuscript entitled "The control effort to steer self-propelled microswimmers depends on their morphology: comparing symmetric spherical vs. asymmetric L-shaped particles" is now accepted for publication in Royal Society Open Science.

on behalf of Dr Kenta Ishimoto (Associate Editor) and Pietro Cicuta (Subject Editor)
openscience@royalsociety.org

Appendix A

The manuscript “The control effort to steer self-propelled microswimmers depends on their morphology: comparing circular and linear movement of symmetric spherical vs. asymmetric L-shaped particles” introduce a information measure they call control effort to measure the effort to complete a task and apply it to two microswimmers in two different environments, a spherical Janus particle and and L shaped swimmer. This measure alone is used to conclude that the L shape are somehow ‘better’.

The navigation and control of microscopic systems is a serious challenge. Improvements on these topics can significantly improve our understanding of bacterial infection, and male fertility and open the doorway for many new technologies. I believe the manuscript in question is genuinely trying to peruse this goal but has fallen short. There is no justification for why control effort is a useful or what it would even mean to the microswimmer community. The manuscript then goes to orchestrate two systems specifically to show “L-shaped particles require less control effort.” Often I found myself asking how is this a fair comparison and noting that when the results didn’t reflect the desired outcome, they were only mentioned briefly. On top of this many of the definitions are missing or appear in random places, and details about the simulations, which would be important to understand, are missing. Finally the discussion is missing many papers related to the design or optimisation of swimmers with special control and completely overlooks the extensive literature on the search behaviour of mircoorganisms. It if for these reasons that I cannot recommend the paper for publication. I have elaborated on these points bellow.

1. Eq. (14) defines the ‘minimal information’ for the motion. The control effort is then defined as this information in a given probability criteria. No where in the text is it explained what this is or why it is a good measure. The only support of this measure we get is “An example: in periodic hopping movements, the minimal information is only about 32 bit/hopping cycle with muscles as compared to 660 bit/cycle with a DC motor Haeufle et al. (2014). Thus, the control effort is much lower for a muscle-driven hopping system.” However this means nothing to me. Is the lower number better because it reflects something or is it purely that the functional has been constructed to do that? I suspect the argument the authors wanted the reader to draw is that ‘biology is better’ and so the lower number is a good sign but this is an evolutionary fallacy as we have no idea what the biological system is optimised for, if at all.

On top of this the information formula requires the system to be discretized in time and barely takes into account the complexity of the underlying control (the logarithm term). Microscopic swimming is however a continuous and complex process. Even though the chemotactic behaviour of *E. coli* is termed run and tumble, the distribution changes continuously in response to the environment. Why should a discretized measure be of any use?

2. Sec. 2.1: The equations for the two swimmers are not formatted for a fair comparison (Eqs. (2),(3),(4),(5)). The Janus particle motion is written in terms of the maximum speed and the L shaped particle is written in terms of a force on the body. Each get some range but no justification to the maxima or the relationship is given. Either both need to have the same max velocity or both have the same force.

Additionally they introduce a second Janus swimmer based on a difference in the rotational diffusion but provide no further comparisons. Why not? What would be a fair equivalence between a sphere and an L shaped particle? There needs to be clear justifications as to the choice of bodies and why you can compare them effectively.

3. Sec.2.2: The L shaped particles are given a much more complicated control strategy. In fact the L shape gets two different strategies while the Janus particle is only provided one. These strategies get more parameters to treat and even allows the L shape to utilise gravity and variable speeds. No attempt to do similar to the Janus particles occurs anywhere within the text.
4. Sec. 3.1 The circle environments chosen is set up for the success of the L shaped swimmer. The circular environment discussed the most within the text is the perfect radius for the L and results in a 100% target when Brownian is off while the Janus gets 0. The discussion of different radii circles are restricted to a single paragraph and note that if the radius gets too large, the L swimmers again fail. How is this an even and fair comparison?
5. The manuscript fails to cite several relevant papers on control and artificial swimmers to the manuscript. For example:
E. Lauga, "Enhanced Diffusion by Reciprocal Swimming," *Phys. Rev. Lett.*, vol. 106, p. 178101, 2011.

T. D. Montenegro-Johnson, “Microtransformers: Controlled microscale navigation with flexible robots,” *Phys. Rev. Fluids*, vol. 3, no. 6, p. 062201, Jun. 2018.

and almost the entire catalogue of A. DeSimone (International School for Advanced Studies, SISSA, Trieste, Italy)

6. The manuscript does not discuss the taxis of microorganisms anywhere. This taxis behaviour is the mechanisms that microorganisms used to reach a target destination and every motile microorganism has been seen to display such behaviour. These processes have been studied at length (look for Pedley or Mitchell). Infact the strategy used for the Janus particle is reminiscent of the run and stop chemotaxis process. Given data on this is so abundant, why is it not even mentioned?
7. Eqs. (4),(5) contain many undefined parameters.
8. Are Eqs. (4),(5) defined in the centre of mobility? If not you get non-Gaussian effects to the diffusion.
9. How was the numerical solutions to Eqs. (2),(3),(4),(5) found, in relation to changing Δt in the information term?
10. “If a particle leaves the channel, it is counted as a fail.” Does this mean they touch the channel edge with part of their body or does their 'centre' need to leave the channel?
11. Figures: Why do the blue lines suddenly end?

Appendix B

Response to Reviews

We refer to the reviewer comments (printed in *italic*), addressing them individually (in **bold**). A compare file between the old and the new version of the manuscript is attached.

Reviewer 1:

The manuscript “The control effort to steer self-propelled microswimmers depends on their morphology: comparing circular and linear movement of symmetric spherical vs. asymmetric L-shaped particles” introduce a information measure they call control effort to measure the effort to complete a task and apply it to two microswimmers in two different environments, a spherical Janus particle and and L shaped swimmer. This measure alone is used to conclude that the L shape are somehow ‘better’. The navigation and control of microscopic systems is a serious challenge. Improvements on these topics can significantly improve our understanding of bacterial infection, and male fertility and open the doorway for many new technologies. I believe the manuscript in question is genuinely trying to peruse this goal but has fallen short. There is no justification for why control effort is a useful or what it would even mean to the microswimmer community. The manuscript then goes to orchestrate two systems specifically to show “L-shaped particles require less control effort.” Often I found myself asking how is this a fair comparison and noting that when the results didn’t reflect the desired outcome, they were only mentioned briefly. On top of this many of the definitions are missing or appear in random places, and details about the simulations, which would be important to understand, are missing. Finally the discussion is missing many papers related to the design or optimisation of swimmers with special control and completely overlooks the extensive literature on the search behaviour of mircoorganisms. It if for these reasons that I cannot recommend the paper for publication. I have elaborated on these points bellow.

Thank you for your comments. It became evident, that central points and the key message behind the manuscript remained unclear. Therefore we revised the introduction, added a more detailed discussion, and changed the structure of the manuscript. We hope that these changes addressed the raised critical points and that they are clear now. (See below for the detailed answers to all of the comments)

1. Eq. (14) defines the ‘minimal information’ for the motion. The control effort is then defined as this information in a given probability criteria. No where in the text is it explained what this is or why it is a good measure. The only support of this measure we get is “An example: in periodic hopping movements, the minimal information is only about

*32 bit/hopping cycle with muscles as compared to 660 bit/cycle with a DC motor Haeufle et al. (2014). Thus, the control effort is much lower for a muscle-driven hopping system.” However this means nothing to me. Is the lower number better because it reflects something or is it purely that the functional has been constructed to do that? I suspect the argument the authors wanted the reader to draw is that ‘biology is better’ and so the lower number is a good sign but this is an evolutionary fallacy as we have no idea what the biological system is optimised for, if at all. On top of this the information formula requires the system to be discretized in time and barely takes into account the complexity of the underlying control (the logarithm term). Microscopic swimming is however a continuous and complex process. Even though the chemotactic behaviour of *E. coli* is termed run and tumble, the distribution changes continuously in response to the environment. Why should a discretized measure be of any use?*

Thank you for this comment. As the meaning of ‘control effort’ is crucial to understand the contribution of the manuscript, we introduced a new section (Sec. 2) to explain this and how control effort is calculated. In a nutshell: with control effort we are able to measure whether the morphology of a system (shape of the particles in this paper) allows to simplify control of a given task. You are indeed correct that biological motion e.g., swimming of bacteria is a “continuous and complex process“ which we cannot directly address with the current implementation of control effort based on discretizing control signals. However, reducing information of a control signal is also possible by other means which can be applied to continuous systems. We briefly elaborate on this in the newly introduced discussion section:

“Currently, our approach to measure control effort relies on reducing the information content by changing the discretization intervals. This limits the direct applicability to other, biologically more plausible and relevant control strategies, e.g., run-and-tumble in chemotactic species which rely on a continuous sampling of a chemical concentration. However, limiting the information content could also be achieved by other approaches, e.g., reducing the signal to noise ratio. We speculate that such an extended approach could also be applied to models of biological microswimmers and may reveal potentially information efficient behavior as one optimization criterion for the biological control strategy.“

Sec. 2.1: The equations for the two swimmers are not formatted for a fair comparison (Eqs. (2),(3),(4),(5)). The Janus particle motion is written in terms of the maximum speed and the L shaped particle is written in terms of a force on the body. Each get some range but no justification to the maxima or the relationship is given. Either both need to have the same max velocity or both have the same force.

We agree with the reviewer that a fair comparison is only possible if the

particles have the same velocity and apologize that this information was a bit hidden before. In short: we calculated the force of the L-shaped particles to match the velocity of the spherical Janus particles $v_{act} = 2.83\mu\text{m/s}$. (Achieved by a force of $F_{max} = 1.47\mu\text{N}$). To make this important information more prominent, we introduced a new subsection *Comparability of different particle shapes*.

Additionally they introduce a second Janus swimmer based on a difference in the rotational diffusion but provide no further comparisons. Why not? What would be a fair equivalence between a sphere and an L shaped particle? There needs to be clear justifications as to the choice of bodies and why you can compare them effectively.

Thank you for this comment. The core idea of the paper was to treat the morphology as the “independent variable“: symmetric vs. asymmetric. Thereby the L-shape particle was chosen as it was reported in the literature to produce complex and partially predictable large-scale motion in contrast to symmetric Janus-particles with a completely stochastic large-scale mean-square displacement ??.

As time to reach the target was one criterion for the evaluation, we adjusted the propulsion mechanism to allow for the same maximum velocity in both models. Additionally, the stochastic Brownian re-orientation of the particles is a crucial factor for the control, as the propulsion mechanism is only turned on if the orientation of the particles is in a desired range (towards the target). Therefore, in addition to the propulsion velocity, we also made the rotational diffusion parameters of the models comparable (see subsection *comparability of different particle shapes*, as well as table 1 for the (rotational) diffusion parameters).

We revised the manuscript to emphasize this in a way that now the calculation for the rotational diffusion parameter of the bigger Janus particles can be found in the new subsection *Comparability of different particle shapes*.

Sec.2.2: The L shaped particles are given a much more complicated control strategy. In fact the L shape gets two different strategies while the Janus particle is only provided one. These strategies get more parameters to treat and even allows the L shape to utilise gravity and variable speeds. No attempt to do similar to the Janus particles occurs anywhere within the text.

The reviewer is correct, the control strategy for the L-shaped particles is more complex and allows to exploit the specific dynamic benefits of the asymmetric particle described in the literature. A transfer of this control scheme to the Janus particles would bring no benefit: varying propulsion velocity

does not change the large-scale movement of the particles (it remains stochastic) and therefore only reduces the distance travelled towards the target. Introducing gravity will add a sedimentation to the particle. As it has no intrinsic dynamics to cope with this, the control would have to be adjusted to constantly also “fight“against sedimentation and therefore reduce the performance. Hence, adding these complexities can only benefit the L-shaped particles and explicitly exploits their reported large-scale motion characteristics which have been reported in the literature.

We revised the manuscript to make this clearer and added a new paragraph at the beginning of section 3.2 Control strategies.

Sec. 3.1 The circle environments chosen is set up for the success of the L shaped swimmer. The circular environment discussed the most within the text is the perfect radius for the L and results in a 100% target when Brownian is off while the Janus gets 0. The discussion of different radii circles are restricted to a single paragraph and note that if the radius gets too large, the L swimmers again fail. How is this an even and fair comparison?

Thank you for this comment. It made clear that we created a general misunderstanding: Our work does not attempt to judge one particle over the other. It rather attempts to show that control effort may be used as a relevant benchmark figure of merit, as it can quantify the contribution of a particles morphology to a specific movement. For this reason, we chose tasks where we could argue for specific expectations. This was in our opinion necessary to validate the method. In the introduction we now specify these in the last two paragraphs and specifically in the following sentences:

“From the experimental trajectories reported in Kümmel et al. (2013); ten Hagen et al. (2014), we expect that (1) the natural rotation of self-propelled L-shaped particles could be exploited for circular movements; (2) the rich behavior in the interaction with gravity in a slightly tilted setup could be exploited for linear movements; (3) that targets exist, which cannot be reached as no control strategy can be found to steer the L-shaped particles in their direction.“

The manuscript fails to cite several relevant papers on control and artificial swimmers to the manuscript. For example:

E. Lauga, “Enhanced Diffusion by Reciprocal Swimming,” Phys. Rev. Lett., vol. 106, p. 178101, 2011.

T. D. Montenegro-Johnson, “Microtransformers: Controlled microscale navigation with flexible robots,” Phys. Rev. Fluids, vol. 3, no. 6, p. 062201, Jun. 2018. and almost the entire catalogue of A. DeSimone (International School for Advanced Studies, SISSA, Trieste, Italy)

We thank the reviewer for pointing out these references. We added them to the introduction and the discussion.

The manuscript does not discuss the taxis of microorganisms anywhere. This taxis behaviour is the mechanisms that microorganisms used to reach a target destination and every motile microorganism has been seen to display such behaviour. These processes have been studied at length (look for Pedley or Mitchell). Infact the strategy used for the Janus particle is reminiscent of the run and stop chemotaxis process. Given data on this is so abundant, why is it not even mentioned?

The reviewer is correct that we did not consider biological behavior here. This is mainly as our current approach cannot directly be applied to continuous chemotactic control. However, to acknowledge this and give an outlook, we added the following sentences to the discussion: “Currently, our approach to measure control effort relies on reducing the information content by changing the discretization intervals. This limits the direct applicability to other, biologically more plausible and relevant control strategies, e.g., run-and-tumble in chemotactic species which rely on a continuous sampling of a chemical concentration. However, limiting the information content could also be achieved by other approaches, e.g., reducing the signal to noise ratio. We speculate that such an extended approach could also be applied to models of biological microswimmers and may reveal potentially information efficient behavior as one optimization criterion for the biological control strategy.”

Eqs. (4),(5) contain many undefined parameters.

The reviewer is correct. We have edited and corrected this.

Are Eqs. (4),(5) defined in the centre of mobility? If not you get non-Gaussian effects to the diffusion.

The equations are defined in the centre of mobility. We added this in the manuscript: “All following equations are defined in the centre of mobility of the particles.”

How was the numerical solutions to Eqs. (2),(3),(4),(5) found, in relation to changing Δt in the information term?

In order to answer this question, it should first be noted that the time steps τ and Δt were changed independently of each other, as they have no dependency on one another. While τ is the integration timestep for the differential equations, Δt is the numerical resolution of the control. We have added a paragraph that goes into more detail about numerical simulation. See quote: “The simulated stochastic motion depends on the diffusion constants D for Janus and L-shaped particles, respectively (Tab.1). For the L-shaped particles with the given dimensions (Tab.1), the diffusion coefficients were obtained experimentally from short-time correlation experiments without gravity and passive sedimentation experiments ten Hagen et al. (2014). The diffusion coefficients for Janus particles with diameter $\sigma = 4.2\mu m$ have also been experimentally determined Haeufle et al. (2016). For comparison with theory these diffusion coefficients had also been calculated by solving the Stokes equation Kümmel et al. (2013) and good agreement with the experimental values has been found ten Hagen et al. (2014). The simulations of motion for both particles were based on a time discrete evaluation of the equations of motion for the two translational degrees of freedom, x and y , and the rotational degree of freedom φ . The differential equations for the particles positions (eqs. 3 and 5) and orientations (eqs. 4 and 6) were solved for constant time intervals of $\tau = 0.5$ s.”

“If a particle leaves the channel, it is counted as a fail.” Does this mean they touch the channel edge with part of their body or does their ‘centre’ need to leave the channel?

The task is considered a fail as soon as the center of the particle leaves the channel. Thank you for this remark. We have changed and clarified this in the text: “If the centre of volume of the particle leaves the corridor, it is counted as a fail.”

Figures: Why do the blue lines suddenly end?

In some figures the blue (also the red and green) lines end. This occurs at every corresponding control time interval at which the arrival probability for the corresponding particle equals zero. We added this in the caption of figure 6, where this effect is particularly prominent and not obvious on first sight. See quote: “However, the comparison of information is made between Janus-particles and L-particles with the highest arrival probability ($N_F = 6$) (subfigure C, lines end, when arrival probability of the particle equals zero).“

Reviewer 2:

The manuscript contains a new approach to quantify the information effort needed to steer a microrobot or an artificial microswimmer to follow a predefined path. The research builds on previous works of the authors and generalizes it to a non-spherical particle.

The generalization might prove useful for the design of general-shaped microswimmers. However, there are a few things which I would like the authors to address before the manuscript is ready for publication.

Thank you for your encouraging review. We hope you find your points adequately addressed.

The control strategies are discussed for a specific model of particles which have anisotropic diffusivity. At the same time, the authors neglect hydrodynamic interactions with walls. Could the authors comment on how much these interactions could modify the presented framework. If a particle was moving close to a wall which would (a) slow down the particle and (b) introduce additional anisotropy of the motion of the particle, e.g. an additional reorientation. These factors would influence the analysed motion in a channel and should be either accounted for or discussed. I fully agree that the presented approach is valid far from boundaries when the motion is affected solely by the externally controlled propulsion which couples to the hydrodynamic anisotropy of the particle but in the presence of nearby walls this description would lose its validity. Perhaps it would be better in this context not to call these predefined trajectories 'channels' but somehow to reflect the fact that there are no physical boundaries which would interact with the swimmer hydrodynamically?

The reviewer is correct: we ignore all physical interactions with the wall which could slow down the particle and change the rotational diffusion of particles near or in contact with a physical wall. Instead, we implement the boundaries only as a non-physical boundary of a channel in the sense of a movement or control goal.

We have revised the corresponding passages in the text and now call them 'target corridor' instead of 'channels', also see quote: "We restrict the movement by limiting the allowed region. If the centre of mass leaves the defined region, further referred to as "target corridor", the attempt is considered a failed navigation. There is no interaction between the particle and a wall, as the corridor is merely a virtual movement constraint."

I was wondering whether this treatment can be generalised to an arbitrary diffusion matrix and thus to a general-shaped particle. Since the control effort is highly specific for the

geometry of the particle. The development of a general framework could be an interesting way forward and I was wondering if the authors could consider a general-shaped particle?

This is an interesting thought, as it may offer the possibility to design a morphology (and diffusion matrix) to optimize for a specific movement goal. In principle, the measure of control effort is general, but in our work the controller rules are particle specific. However, the general on-off steering strategy towards a desired goal could be used as the generalised controller and be therefore used to compare particles with arbitrary diffusion matrices. We added this thought to the discussion section (second to last paragraph).

There is a number of systems in which external factors can switch on/off active propulsion. Examples are: Feng et al Appl Phys Rev (2021) Vutukuri et al Nature Comm (2020) Dong et al ACS Nano (2016) Jiang et al Phys Rev Lett (2010) and others in literature. It would be beneficial if the authors could discuss the applicability of their findings to such systems.

We thank the reviewer for pointing this out. We mention the general applicability of the approach to other systems now at the beginning of the methods and added the following to the Discussion:

“The basis for our models were active synthesized particles with a photophoretic self-propulsion mechanism Volpe et al. (2011); Kümmel et al. (2013); Bechinger et al. (2016). However, the control approach (on-off strategy) and the evaluation (control effort) would also be applicable to systems with other propulsion mechanisms which allow online on/off switching of the active propulsion Dong et al. (2016); Feng et al. (2021); Jiang et al. (2010); Vutukuri et al. (2020). “

Could the Authors comment more on whether L-shaped particles with the ability to switch their propulsion on/off have been synthesized?

Indeed, they have been synthesized and we now emphasize this in the introduction: “From the experimental trajectories reported in Kümmel et al. (2013); ten Hagen et al. (2014), we expect that...”

A couple of minor points:

- the Authors should use uniform notation for the decimal point (not comma) throughout the paper.

Thank you for pointing this out. We have revised the paper in respect thereof.

- in the introduction, when listing papers involving control strategies, the Authors could specifically comment on what these strategies are based on.

Thank you for this comment. We state in the introduction: “The control strategy is to wait until rotational diffusion randomly orients the particle towards the target. Only then propulsion is activated allowing for a simple navigation of the particle Qian et al. (2013); Bregulla et al. (2014).“

In Table 1 and above eq. (6), the parameter ℓ seems not to be defined (or at least I could not find the definition. It would be helpful if the authors could clarify this point

The reviewer is correct. We included a definition.

the symbol T is used both for temperature and time required for movement. Also, T is not defined in Eq. (14).

We revised this and clarified the symbols and definitions.

The last sentence of Sec. 2 is not clear.

We revised the sentence to: “In this sense, control effort is the minimal information required to navigate the particles to the target within the target corridor limits with the constraint of a desired arrival probability“

Literatur

- B. ten Hagen, F. Kümmel, R. Wittkowski, D. Takagi, H. Löwen, and C. Bechinger, Nat. Commun. **5**, 4829 (2014), ISSN 2041-1723, URL <http://www.ncbi.nlm.nih.gov/pubmed/25234416><http://www.nature.com/doi/10.1038/ncomms5829>.
- D. F. B. Haeufle, T. Bäuerle, J. Steiner, L. Bremicker, S. Schmitt, and C. Bechinger, Phys. Rev. E **94**, 1 (2016), URL <http://arxiv.org/abs/1607.03266>.
- F. Kümmel, B. Ten Hagen, R. Wittkowski, I. Buttinoni, R. Eichhorn, G. Volpe, H. Löwen, and C. Bechinger, Phys. Rev. Lett. **110**, 1 (2013), ISSN 00319007.
- G. Volpe, I. Buttinoni, D. Vogt, H.-J. Kuemmerer, and C. Bechinger, p. 4 (2011), ISSN 1744-683X, [1104.3203](https://doi.org/10.1038/1104.3203), URL <http://arxiv.org/abs/1104.3203>.

- C. Bechinger, R. Di Leonardo, H. Löwen, C. Reichhardt, G. Volpe, and G. Volpe, *Rev. Mod. Phys.* **88**, 045006 (2016), URL <https://link.aps.org/doi/10.1103/RevModPhys.88.045006>.
- R. Dong, Q. Zhang, W. Gao, A. Pei, and B. Ren, *ACS Nano* **10**, 839 (2016), ISSN 1936086X.
- Y. Feng, Y. Yuan, J. Wan, C. Yang, X. Hao, Z. Gao, M. Luo, and J. Guan, *Applied Physics Reviews* **8**, 011406 (2021), ISSN 1931-9401, URL <https://aip.scitation.org/doi/10.1063/5.0029060>.
- H. R. Jiang, N. Yoshinaga, and M. Sano, *Physical Review Letters* **105**, 1 (2010), ISSN 00319007, [1010.0470](https://doi.org/10.1073/prl.105.01.01).
- H. R. Vutukuri, M. Lisicki, E. Lauga, and J. Vermant, *Nature Communications* **11**, 1 (2020), ISSN 20411723, [2012.00107](https://doi.org/10.1038/s41467-020-15764-1), URL <http://dx.doi.org/10.1038/s41467-020-15764-1>.

Appendix C

IMSB, University of Stuttgart
Nobelstr. 15
70569 Stuttgart
Germany

August 29, 2021

Dear Professor Cicuta, dear Dr Kenta Ishimoto,

we want to thank you and both reviewers for the given input, comments and criticism of our manuscript and are glad to hear, that our revision made the aim of the manuscript „**The control effort to stir self-propelled microswimmers depends on their morphology: comparing symmetric spherical vs. asymmetric L-shaped particles**“ clear.

We are also pleased to hear that our work is considered an interesting contribution and is now recommended for publication.

Please find attached our final corrected version of the manuscript. We proofread the manuscript again and corrected the remaining typos, as well as changed the bibliography style to vancouver style. The file tracking the made changes was uploaded.

Thank you again for your time and assistance, as well as leading the manuscript through the whole review process.

Sincerely,
Julia Riede